# Seasons in Drift: A Long-Term Thermal Imaging Dataset for Studying Concept Drift

**Ivan Nikolov**[1], **Mark P. Philipsen**[1], **Jinsong Liu**[1], **Jacob V. Dueholm**[1], **Anders S. Johansen**[1], **Kamal Nasrollahi**[1,2], and **Thomas B. Moeslund**[1]

[1] Visual Analysis and Perception Lab, Aalborg University, Aalborg, Denmark
[2] Research, Milestone Systems, Brøndby, Denmark

## Abstract

The time dimension of datasets and long-term performance of machine learning models have received little attention. With extended deployments in the wild, models are bound to encounter novel scenarios and concept drift that cannot be accounted for during development and training. In order for long-term patterns and cycles to appear in datasets, the datasets must cover long periods of time. Since this is rarely the case, it is difficult to explore how computer vision algorithms cope with changes in data distribution occurring across long-term cycles such as seasons. Video surveillance is an application area clearly affected by concept drift. For this reason we publish the Long-term Thermal Drift (LTD) dataset. LTD consists of thermal surveillance imaging from a single location across 8 months. Along with thermal images we provide relevant metadata such as weather, the day/night cycle and scene activity. In this paper we use the metadata for in-depth analysis of the causal and correlational relationships between environmental variables and the performance of selected computer vision algorithms used for anomaly and object detection. Long-term performance is shown to be most correlated with temperature, humidity, the day/night cycle and scene activity level. This suggests that the coverage of these variables should be prioritised when building datasets for similar applications. As a baseline, we propose to mitigate the impact of concept drift by first detecting points in time where drift occurs. At this point we collect additional data that is used to retraining the models. This improves later performance by an average of 25% across all tested algorithms.

## 1 Introduction

Once computer vision algorithms step outside the lab and are deployed in real-life outdoor applications, their performance tends to drop significantly due to conditions changing over time, i.e. concept drift [90, 24, 85]. Concept drift can materialize as gradual, recurring or sudden changes in the visual representation of the scene. Existing datasets, in general, favour coverage of multiple locations [32, 75] for short periods of time [46, 45, 83]. Such datasets are ill suited for exploring long-term effects such as concept drift and algorithms developed on their basis are unlikely to show robustness to long-term phenomena. Research studying concept drift [28, 55], uses synthetic datasets or datasets augmented in order to introduce drift. This does not necessarily completely represent real-world concept drift.

Our work presents a novel real-world dataset covering the 8 months from January to August. This time span means that the dataset encompasses a wide range of weather conditions, human activity, seasonal transitions, and recurring cycles such as weekdays, weekends, mornings and evenings. Along with the thermal images, timestamped metadata has been gathered. The metadata includes

35th Conference on Neural Information Processing Systems (NeurIPS 2021) Track on Datasets and Benchmarks.

weather data such as temperature, humidity, precipitation, etc. as well as metrics for scene activity level. We use the dataset to study concept drift by exploring contributing factors and demonstrating their effects on algorithmic performance. By publishing the dataset, we seeks to aid the community in evaluating exiting algorithms against a long-term benchmark and in the development of algorithms that show greater robustness to long-term phenomena.

To explore the dataset, two common tasks are chosen, namely anomaly and people detection. These tasks tend to suffer strong performance degradation when exposed to long-term concept drift [77]. Object detection in general or detecting people in particular is a fundamental task involved in many use cases such as autonomous driving [86, 10, 8], tracking [6, 67, 73, 19] and re-identification [40, 41, 26]. Common for many of the use cases is the application of object detection in unconstrained environments and across long spans of time. Anomaly detection, where the goal is to detect unusual behavioral patterns, is another task that is exposed to concept drift. These algorithms must be able to distinguish irrelevant changes due to e.g. concept drift from emergencies such as burglaries or assaults [75], car accidents [39], loitering and suspicious behaviour [89], indoor [27] and outdoor[15, 36, 43] falls.

We select representative algorithms for each task and evaluate their performance across time and in relation to environmental factors. As expected, all models exhibit performance degradation, as the test data diverges from the training set. Temperature and humidity proves to influence the models the most, followed by the change between day and night and the activity level of the scene. On the other hand, variation in precipitation and wind do not influence the performance of the models. In general, methods that learn from solving tasks that consider the entirety of the image are likely to be less impacted by drift, compared to methods that consider small regions or individual pixels [76]. An example could be object detectors vs. autoencoders, where something like brightness is likely to impact the autoencoder's reconstruction significantly, but won't effect the class or position of objects. By including both autoencoders and object detectors we ensure that both ends of this spectrum are covered in our analysis.

Finally, a baseline algorithm is presented to reduce the consequences of concept drift. This algorithm provides additional training data from points in time where concept drift is detected. This baseline is intended to encourage researchers to develop other methods of reducing the impact of concept drift. We believe that our findings on this novel dataset generalize to other environments and use cases, as well as other modalities and therefore will be an example to follow for future definition and collection of datasets. This in turn will help the community getting closer to deploying long-term computer vision algorithms for real-life outdoor applications. The main contributions of this paper can be summarized as follows:

- The Long-term Thermal Drift (LTD) dataset - the longest-spanning systematically collected thermal dataset comprised of 8 months of video data, containing both timestamp and weather condition metadata;

- In-depth analysis of the correlational and causal relationships between the performance of models and environmental factors;

- A baseline algorithm for reducing the effects of concept drift.

## 2 Related Work

### 2.1 Concept Drift Detection

As many systems need to be deployed and work stably for long periods of time and with input data which can change both gradually and suddenly, the presence of drift and ways to deal with it is a topic that has been widely studied. In computer vision it is normally studied by either focusing on specific real-world use cases or synthetically augmenting existing datasets. Real-world cases can be taken from egocentric video [53] or industrial inspection [52]. These cases present both examples of the problem and detection methods, but have limited use outside of the specific environments. Augmented versions of popular datasets such as MNIST and CIFAR can also be used. The works by [55] and [61] focus on methods for detecting data shifts using differences between the training and testing data, utilizing dimensionality reduction and statistical tests like Maximum Mean Discrepancy and Kolmogorov-Smirnov test. The benefit of using synthetically augmented data for testing is that different types of shifts can easily be simulated - from gradual drift to adversarial attacks [28]. But

Table 1: Existing urban computer vision stationary and changing location datasets. The *Location* can be either changing denoting moving camera like the ones on self-driving cars or stationary like on surveillance cameras. The *Type* of the datasets can be either RGB, thermal or LiDAR, the *Duration* is the size of the dataset in hours, the *Period* is the capturing time span and the *Metadata* is any additional information

| Name | Year | Location | Type | Duration [hours] | Period | Metadata |
|---|---|---|---|---|---|---|
| KAIST [32] | 2015 | Changing | RGB/Thermal | 43.41 | - | - |
| CVC-14 [20] | 2016 | Changing | RGB/Thermal | 11.8 | - | - |
| Oxford RobotCar [48] | 2017 | Changing | RGB/LiDAR | - | 1 year | GPS, IMU, Day/Night, Weather |
| Aachen Day-Night [70] | 2018 | Changing | RGB | - | - | GPS, Day/Night, Weather |
| Gated2Depth [23] | 2019 | Changing | RGB/LiDAR | - | - | GPS, IMU, Day/Night, Weather |
| Dark Zurich [68] | 2019 | Changing | RGB | - | - | GPS, Day/Night |
| ACDC [69] | 2020 | Changing | RGB | - | several days | GPS, Weather |
| Ford AV [1] | 2020 | Changing | RGB/LiDAR | - | 1 year | GPS, IMU Day/Night, Weather, Time |
| Bdd100k [87] | 2020 | Changing | RGB | - | - | Weather, Time |
| UCSD [49] | 2010 | Stationary | RGB | 3.1 | - | - |
| Caltech Pedestrian [13] | 2011 | Stationary | RGB | 10 | - | - |
| VIRAT [54] | 2011 | Stationary | RGB | 29 | - | - |
| Avenue [46] | 2013 | Stationary | RGB | 0.5 | - | - |
| ShanghaiTech Campus [45] | 2018 | Stationary | RGB | 3.6 | - | - |
| Surveillance Videos [75] | 2018 | Stationary | RGB | 128 | - | - |
| Street Scene [62] | 2020 | Stationary | RGB | 4 | 2 summers | - |
| ADOC [60] | 2020 | Stationary | RGB | 24 | 1 day | - |
| AU-AIR [5] | 2020 | Stationary | RGB | 2 | - | Time, Positions |
| MEVA [12] | 2021 | Stationary | RGB/Thermal | 144 | 3 weeks | GPS, Time |
| **LTD (Our)** | **2021** | **Stationary** | **Thermal** | **298** | **8 months** | **GPS, Day/Night, Weather, Time** |

these simulated shifts do not always correspond to real-world ones. Some more robust methods also exist [77], aimed at using real-world drift in wider variaty of use cases. The need for more research into concept drift, paired with a long-term real-world dataset is evident, as the effects from it can limit long term deployment of vision systems [72, 2].

## 2.2 Datasets

We can separate previous work roughly in two types of use cases - datasets that contain a scenes from a stationary location, like the ones captured from CCTV and surveillance cameras and datasets with constantly changing locations, like the ones specifically directed towards autonomous cars, robots and human egocentric footage. The two types of datasets are used for different tasks, like vehicle and pedestrian detection and environmental segmentation for changing datasets [32, 87, 1] and pedestrian tracking and anomaly detection for stationary ones [45, 62, 12]. The changing datasets also benefit from more diverse data coming from different sensors, compared to more image based stationary datasets. Our proposed LTD dataset is directed towards advancing the state-of-the-art in stationary location outdoor urban datasets by providing a longer duration, larger variation and rich metadata. A comparison in Table 1 shows how the dataset stacks against previous work.

Datasets used for autonomous driving with changing locations [87, 70, 23, 1], which contain multiple modalities like LiDARs, RGB, depth cameras, as well as GPS and IMU data. They also contain data with longer duration from multiple days [69] to a whole year [48]. These datasets also focus on presenting adverse weather conditions, which can be used for domain adaptation and making autonomous driving and robotics application more robust [68, 1, 69].Thermal datasets are less prevalent but still widely used [17, 20]. These moving location car datasets normally do not contain explicit information of their duration, as they are captured from many cars and the data is sampled.

On the other hand stationary location datasets do not contain any information about the period over which they were collected. This combined with the relative short duration of many of the widely used datasets ([49, 45, 13, 44]) makes it impossible for them to be used for studying long-term effects on deployed machine learning solutions. The duration of some of these datasets is taken from the research presented in [60]. Some larger datasets are gathered from internet videos [75], which lack the needed continuity for testing gradual concept drift in the data. More recent datasets have been produced with the goal to capture larger variations in the environments [12, 60], but with a limited scope. The lack of metadata is another problem, limiting the study of factors causing concept drift, as only some of the investigated datasets provide insufficient metadata [5, 12, 66]. Most of

the investigated datasets focus on RGB data, with only some containing both RGB and thermal data [32, 12]. However, thermal imaging is better at preserving preserving people's anonymity as it does not capture facial and body detail. This removes the need for post-processing like blurring or pixelating faces to protect personal data [88, 47, 37], which is a crucial requirement for complying with the European general data protection regulations (GDPR).The thermal imaging market has seen significant growth [14] and is forecast to expand even more in the following years [65, 34], which makes it necessary for long-term public thermal datasets to be easily accessible

## 3    The Long-term Thermal Drift (LTD) Dataset

To address the gaps seen in the stationary surveillance state-of-the-art and to leverage the need for more thermal data, a new dataset is proposed. It consists of thermal videos with resolution $288 \times 384$ captured through the period of **8 months** using a Hikvision DS-2TD2235D-25/50 thermal camera [30]. The camera is a long wavelength infrared (LWIR) unit, capturing wavelengths between 8 and 14 $\mu m$. Raw data is captured through the day and saved in a mp4 format as 8-bit uncalibrated grayscale videos. A pre-processing algorithm is then run through the data. It first cuts the raw files into days starting from $00:00$ and separates them into folders. Each folder is timestamped with the year, month and day timestamp. The videos for each day are then cut into **2-minute** clips selected every 30 minutes through the day, for a total of **298 hours**. These videos are additionally timestamped with hour and minute timestamp. The starting point of the data is May 2020 until September 2020, together with a second part from January 2021, up until May 2021. This gives the data a large weather variation through the winter, spring and summer seasons. The images were taken on the harbor front in Aalborg, Denmark. The approximate longitude and latitude coordinates are given as $(9.9217, 57.0488)$. We provide the dataset - **https://www.kaggle.com/ivannikolov/longterm-thermal-drift-dataset**, together with the code to extract the necessary data and to reproduce the experimental pipeline **https://github.com/IvanNik17/Seasonal-Changes-in-Thermal-Surveillance-Imaging**.

Some examples of seasonal and day and night variation of the captured data, together with weather and human activity variation can be seen in Figure 1. These large variations, together with a total size almost twice as large as other datasets in Section 2.2, allows for studying the effects of concept drift on trained models.

Seasons

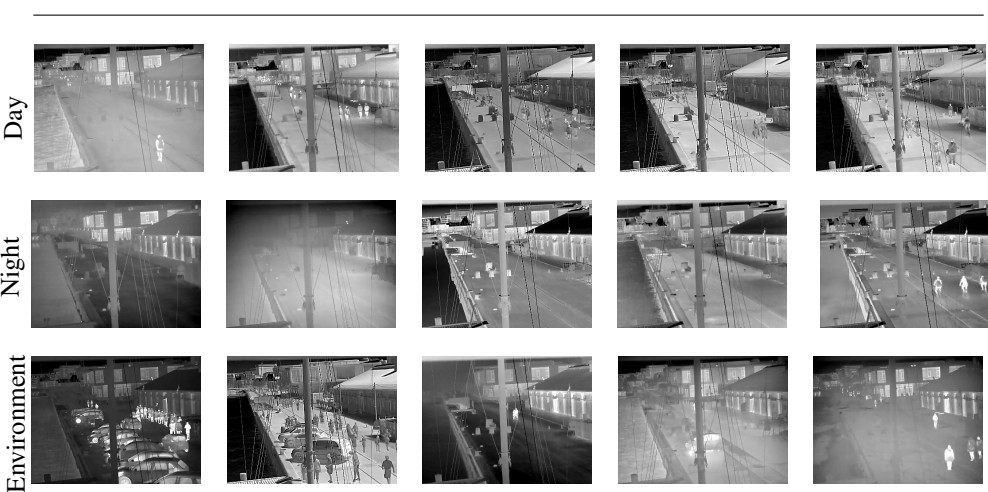

Figure 1: Examples of extreme changes in the image data contained in the proposed dataset. From left to right the day and night rows show example changes from data of February, March, April, June and August. The third row shows changes based on weather conditions and human activity.

Figure 1 depicts issues stemming from the natural thermal data concept drift, such as grayscale inversion in the background and people in different seasons, view limitation and reflections caused by weather like fog, rain, snow, view cluttering from multiple people and vehicles.

Table 2: Average metadata for each month. From left - temperature, humidity, precipitation, dew point, wind direction, wind speed, sun radiation and minutes of sunshine in a 10-minute interval.

| | Temp. [$^\circ C$] | Hum. [%] | Precip. [$kg/m^2$] | Dew P. [$^\circ C$] | Wind Dir. [$degrees$] | Wind Sp. [$m/s$] | Sun Rad. [$W/m^2$] | Sun [$min$] |
|---|---|---|---|---|---|---|---|---|
| Jan. | -0.48 | 90.10 | 0.01 | -1.96 | 161.91 | 2.58 | 23.97 | 0.90 |
| Feb. | -0.54 | 85.15 | 0.01 | -2.83 | 131.00 | 2.95 | 51.12 | 1.42 |
| Mar. | 3.75 | 83.61 | 0.01 | 0.93 | 218.80 | 3.58 | 99.35 | 1.85 |
| Apr. | 4.47 | 97.25 | 0.13 | 4.10 | 126.50 | 2.97 | 67.31 | 2.23 |
| May | 10.74 | 75.46 | 0.01 | 6.07 | 217.32 | 3.04 | 256.76 | 3.66 |
| June | 16.36 | 71.46 | 0.01 | 10.57 | 151.27 | 2.37 | 256.46 | 3.63 |
| July | 12.91 | 75.32 | 0.01 | 8.46 | 268.15 | 3.97 | 270.17 | 3.62 |
| Aug. | 16.93 | 79.17 | 0.02 | 12.69 | 163.18 | 2.08 | 197.86 | 3.15 |

## 3.1 Metadata Analysis

Besides video data we also provide metadata in the form of weather data, gathered using the open source Danish Meteorological Institute (DMI) weather API [33] in 10-minute intervals. The selected properties are - temperature, measured in [$^\circ C$], relative humidity percentage measured 2m over terrain, accumulated precipitation in [$kg/m^2$], dew point temperature in [$^\circ C$] measured 2m over terrain, wind direction in degrees orientation, wind speed in [$m/s$], both measured 10m over terrain, mean sun radiation in [$W/m^2$] and minutes of sunshine in the measured interval. These properties are selected, as it is speculated that they would be useful to explain changes in the captured image data. An overview of the average weather metadata measurements of the dataset can be seen in Table 2. Temperature and relative humidity have been shown to affect thermal cameras, when detecting surface defects in concrete structures [80], measuring skin temperature changes on athletes [35], getting accurate readings for volcanology [3] and inspecting food [21]. Precipitation and dew point temperature can indicate the presence of rain, fog or high moisture and condensation. These can increase attenuation of infrared light and change the produced camera response [4, 11]. The build-up of moisture can create puddles in the images, which would change the scene reflectivity and reflected temperature [7]. The sun radiation and amount of sunshine can affect the captured images by rapidly changing the intensity of the infrared light. Finally wind speed and direction can cause movement of background parts of the scene like water ripples, ropes, etc., as well as movement of the camera itself.

## 4 Long-term Performance Experiment

We study the effects of concept drift on six machine learning models - two autoencoders, two object detectors and anomaly detectors. For these experiments only weather parameters not found to have significant correlation to other parameters are considered, namely - temperature, humidity, wind speed, wind direction and precipitation. More information on the correlation between weather parameters is given in the Appendix.

## 4.1 Data Selection Protocol

In order to keep the experiments and labelling effort manageable, samples across the full data set are selected based on the following protocol. This is done to minimize the number of frames and maximize the variation covered by the selection. For the sampling temperature metadata is used, as it is proven to directly correlate with changes to thermal images [80, 35, 21]. The protocol can be summarized as follows:

1. Every **2-minute** clip in the dataset is sampled with a frequency of one frame per second, resulting in **120 frames per clip**;

2. Based on the temperature metadata, we select a cold month for the training set and another cold month, a median temperature one, and a warm month for the test set;

3. The training set exists in three variants: coldest day 13th of February, the corresponding week 13-20 of February, and the entirety of February;

4. The test sets consist of data from January (similar cold month), April (month with median temperature), and August (warmest month).

From each of the thus created subsets, a greedy furthest point sampling is used for selecting frames. The frames for each day are sampled by calculating the farthest distances in the 2D feature space of the frame numbering and the temperature. A visual example of the sampling can be seen in the Appendix. The amounts of selected samples vary for the training data depending on the used algorithm. This is further discussed in the next sections.

## 4.2 Tested Models

Six deep learning models are tested. All six are originally designed to work with RGB data, so their input channel is reduced from 3 to 1, corresponding to a change to the grayscale thermal data. No additional changes were made, as the focus of the paper is not algorithm performance but change in performance over time.

Two of tested models are autoencoders, as representatives for dimensional reduction, noise removal, concept drift detection and anomaly detection methods. Autoencoders are well suited for researching concept drift in long-term datasets, as their reconstruction performance is inherently tightly connected to the training data. The first autoencoder follows a simple fully convolutional architecture with symmetric 5-layer encoder and decoder. The implementation is based on the autoencoder used in a previous work [43]. It is theorized that its simplicity will make it sensitive to concept drift in the input data. The second autoencoder is the latest version of the Vector Quantised Variational Autoencoder (VQVAE2) [63]. This autoencoder uses collections of multi-scale hierarchical discrete tensors, called codebooks, to map its latent space. This gives it more robustness compared to regular autoencoders. The VQVAE2 implementation used here is closely based on [50]. Both autoencoders are trained for 200 epochs.

Two versions of the anomaly detector method MNAD [57] are also tested. They extend traditional autoencoders, by introducing memory-guided normality detection. We look at the typical reconstruction based comparison (MNAD_recon), as well as the prediction approach (MNAD_pred), using the preceding four consecutive frames to predict the future frame. The backbone consists of the U-Net structure, without skip-connections for the MNAD_recon variant. In between the encoder and decoder of U-Net is a memory module, storing prototypical events, concatenated with the original encoder output. The memory is primarily learned during training, but also updates during testing. Both versions are trained for 100 epochs.

Lastly two supervised object detectors are also tested - the YOLOv5 and Faster R-CNN[64]. The chosen hyperparameters for YOLOv5 remain the same as the work in [82], except that the initial learning rate is set to 0.00075 and trained for 200 epochs. The Faster R-CNN is trained for 200 epochs as well with SGD, with initial learning rate set as 0.005, the weight decay as 0.005 and the momentum kept at 0.9. Both object detectors have previously been successfully applied to outdoor thermal imaging [38, 31, 9, 18].

The autoencoders are trained on a NVIDIA GTX1070 Super, the anomaly detectors on a NVIDIA RTX3080 and the object detectors on a NVIDIA RTX2080Ti.

## 4.3 Drift Algorithmic Performance Analysis

This experiment aims to see how the performance of the selected algorithms changes depending on the variation of the training data.

The training sets for the autoencoders and the anomaly detectors contain 5000 frames per subset, sampled using the method discussed in subsection 4.1, where 20% are used for validation. Performance is reported as the average MSE across every image in each of the three test sets. The performance of the two autoencoders and anomaly detectors is listed in Table 3. We can see that the MSE for the CAE, VQVAE2 and MNAD_recon increases the farther away the test data goes from the training data. It can also be seen that the larger temporal pool provided for sampling for the weekly and monthly training data helps with keeping the MSE lower through the different months. The MNAD_pred is the only model keeping a consistent performance through the months without any noticeable drift. This is most likely due to the U-Net skip connections being able to reconstruct the background scene with a very low reconstruction error.

For the object detectors, because of the necessary data-labeling a smaller number of images are used for training and testing - both having 100 frames per subset. In addition to these a validation

set comprising of 51 images evenly sampled from a previous annotated dataset [43] collected in February 2020 is used. All of the subsets are annotated with bounding boxes around people seen in each frame using the LabelImg open source program [81]. The annotations are also part of the LTD dataset. Since the performance of object detector is based on detected bounding boxes, mAP is used to evaluate it. The performance of the object detectors is given in Table 4. The accuracy of both object detectors, drastically drops in the month of April. To prevent overfitting the smaller amount of training data, we observe the validation and test loss.

As a conclusion from the performance analysis the higher variation provided by sampling from the week and month data, has been translated to better and more stable models in all the tested models. We can still see the effects of the seasonal drift, so additional analysis will be provided in the following sections.

Table 3: Results are reported as the average of the MSE across every frame in the test set. Higher results show worse performance.

| Methods | Train Feb. | Test Jan. | Apr. | Aug. |
|---|---|---|---|---|
| CAE | Day 5k | 0.0096 | 0.0202 | 0.0242 |
| | Week 5k | 0.0061 | 0.0167 | 0.0212 |
| | Month 5k | 0.0042 | 0.0109 | 0.0147 |
| VQVAE2 | Day 5k | 0.0051 | 0.0072 | 0.0068 |
| | Week 5k | 0.0039 | 0.0066 | 0.0061 |
| | Month 5k | 0.0021 | 0.0039 | 0.0035 |
| MNAD Recon. | Day 5k | 0.0028 | 0.0057 | 0.0069 |
| | Week 5k | 0.0065 | 0.0066 | 0.0062 |
| | Month 5k | 0.0015 | 0.0041 | 0.0048 |
| MNAD Pred. | Day 5k | 0.0008 | 0.0007 | 0.0009 |
| | Week 5k | 0.0007 | 0.0006 | 0.0007 |
| | Month 5k | 0.0007 | 0.0006 | 0.0007 |

Table 4: Results are reported as the $mAP_{50}$ across every frame in the test set. Lower results show worse performance.

| Method | Train Feb. | Test Jan. | Apr. | Aug. |
|---|---|---|---|---|
| YOLOv5 | Day 100 | 0.8010 | 0.5390 | 0.5240 |
| | Week 100 | 0.7940 | 0.4540 | 0.4860 |
| | Month 100 | 0.7930 | 0.4860 | 0.4830 |
| Faster R-CNN | Day 100 | 0.6760 | 0.3230 | 0.3370 |
| | Week 100 | 0.6740 | 0.2790 | 0.3060 |
| | Month 100 | 0.6400 | 0.2560 | 0.3180 |

## 5 Drift Analysis

In this section we look at the possible relations between the observed model performance drift and the changes in the captured metadata. Looking through the data examples given in Figure 1, two main visual change types are identified - seasonal and day/night. These types can be caused by either changes in the weather conditions, the human activity or a combination between the two. The relation between the model performance metrics and metadata features representing these changes is analysed. As discussed in section 3.1, we choose temperature, humidity, precipitation, wind direction and wind speed as weather data features. For analysing the day/night changes the timestamp data is used to calculate hours of the day, as well as to calculate the sunrise and sunset times [74, 51]. To quantify the activity in the scene the difference between each testing frame and the previous frame from the main dataset is calculated. The mean value from this difference is selected. To focus only on scene activity everything in the background that moves like the waterfront and the visible ropes and masts is masked out. More information on this can be found in the Appendix.

We choose to use the results only from the models trained on the monthly February data, for easier visualization. The correlation between each of these features and the measured performance metric for each of the methods is first calculated. For the autoencoders and anomaly detectors this is the MSE, while for the object detectors we calculate the F1-score from all images containing people, as it gives a good overview of the precision and recall of the models. Both the basic Pearson's correlation, as well as the more sensitive to non-linear relations Distance correlation [78, 16] are calculated. The statistical significance p-values are also calculated with threshold at 0.05. The calculated correlation $r$ values are given in Table 5, where those with p-values below the threshold are shown in red.

From Table 5 it can be seen that temperature and humidity have both the largest correlation values to most of the metrics, as well as the most consistently statistically significant results, followed by the scene activity and day/night features. We focus on these four features in the following analysis.

To get a better understanding of not only the correlational, but also causal relations between the models' performance metrics and the chosen features, we look at the Granger causality test [22].

Table 5: Correlation between the model's measured performance values MSE and F1-score and the weather, time and scene activity features. Two correlation measures are used - Pearson's (P.C.) and Distance (D.C.) correlation. Measures which do not meet the statistical significance threshold of their p-values are shown in red and marked ✗. The Day/Night features is specified as D./N.

| | Measure | Temp. | Hum. | Wind Dir. | Wind Sp. | Precip. | Activ. | D./N. | Hour |
|---|---|---|---|---|---|---|---|---|---|
| CAE - MSE | P. C. | 0.679 | 0.636 | 0.018 ✗ | 0.157 | 0.109 ✗ | 0.270 | 0.545 | 0.166 |
| | D. C. | 0.682 | 0.588 | 0.158 | 0.170 | 0.126 ✗ | 0.291 | 0.538 | 0.287 |
| VQVAE2 - MSE | P. C. | 0.381 | 0.690 | 0.001 ✗ | 0.194 | 0.172 | 0.217 | 0.403 | 0.124 |
| | D. C. | 0.347 | 0.639 | 0.174 | 0.201 | 0.224 | 0.217 | 0.382 | 0.213 |
| MNAD Recon. - MSE | P. C. | 0.607 | 0.672 | 0.016 ✗ | 0.173 | 0.126 | 0.220 | 0.509 | 0.156 |
| | D. C. | 0.617 | 0.629 | 0.188 | 0.177 | 0.155 | 0.252 | 0.501 | 0.273 |
| MNAD Pred. - MSE | P. C. | 0.107 ✗ | 0.277 | 0.064 ✗ | 0.152 | 0.072 ✗ | 0.677 | 0.369 | 0.137 |
| | D. C. | 0.231 | 0.348 | 0.154 | 0.172 | 0.086 ✗ | 0.665 | 0.462 | 0.312 |
| YOLOv5 - F1-score | P. C. | 0.261 | 0.258 | 0.102 ✗ | 0.011 ✗ | 0.096 ✗ | 0.124 ✗ | 0.047 ✗ | 0.009 ✗ |
| | D. C. | 0.293 | 0.283 | 0.146 ✗ | 0.094 ✗ | 0.135 ✗ | 0.255 | 0.113 ✗ | 0.174 ✗ |
| Faster R-CNN - F1-score | P. C. | 0.354 | 0.456 | 0.115 ✗ | 0.135 ✗ | 0.0124 ✗ | 0.199 | 0.147 | 0.001 ✗ |
| | D. C. | 0.334 | 0.460 | 0.228 | 0.149 ✗ | 0.065 ✗ | 0.231 | 0.163 | 0.118 ✗ |

The test only guarantees a predictive causality between variables, but would be able to point out any possible connections. The Granger causality tests the null hypothesis that the past values of one variable do not cause another. The p-value threshold is set to 0.05, below that the null hypothesis can be rejected, with the conclusion that there is a predictive causality between the variables. As the normal Granger causality test as presented in [71] is used on data with linear relations, we also use the more robust non-linear Neural Granger test [79]. Two best performing versions are used, based on long-short term memory networks (LSTM) and multi-level perceptron (MLP). Both models were trained using proximal gradient descent [56], with $\lambda = 0.002$, ridge regression coefficient $0.01$ and learning rate of $0.005$. The results from the Granger causality tests are given in Table 6, where cells shown with green indicate a statistically significant presence of Granger causality and the ones with red - no presence.

Table 6: Results from calculating linear and non-linear (LSTM and MLP) Granger causality tests. The cells marked with ✓ show positive predictive causality, while cells marked with ✗ show no significant causality.

| | Temp. | | | Hum. | | | Activ. | | | D./N. | | |
|---|---|---|---|---|---|---|---|---|---|---|---|---|
| | Basic | LSTM | MLP | Basic | LSTM | MLP | Basic | LSTM | MLP | Basic | LSTM | MLP |
| CAE - MSE | ✓ | ✓ | ✓ | ✓ | ✓ | ✗ | ✗ | ✗ | ✗ | ✓ | ✓ | ✓ |
| VQVAE2 - MSE | ✓ | ✓ | ✓ | ✓ | ✓ | ✗ | ✗ | ✗ | ✗ | ✓ | ✓ | ✓ |
| MNAD Recon. - MSE | ✓ | ✓ | ✓ | ✓ | ✓ | ✗ | ✗ | ✗ | ✗ | ✓ | ✓ | ✓ |
| MNAD Pred. - MSE | ✓ | ✓ | ✗ | ✗ | ✗ | ✗ | ✗ | ✓ | ✗ | ✓ | ✓ | ✓ |
| YOLOv5 - F1-score | ✓ | ✗ | ✗ | ✓ | ✗ | ✓ | ✗ | ✗ | ✗ | ✗ | ✗ | ✗ |
| Faster R-CNN - F1-score | ✗ | ✗ | ✗ | ✗ | ✓ | ✗ | ✗ | ✗ | ✗ | ✓ | ✓ | ✓ |

The results show that the human activity has no predictive causality towards the performance of the models, which combined with the results from the correlation analysis, can point towards a second-hand relation. Our hypothesis is that the change in weather conditions and the day/night cycle are related to the change in human activity. From the other features, temperature has stronger predictive causality towards the autoencoders and anomaly detectors, while humidity and the day/night cycle have a more balanced predictive causality.

Figure 2 shows the relationship between the features and the model metrics. As a processing step before plotting the temperature and humidity they are first smoothed using a mean filter with a kernel size of 20 and then the MSE is normalized between 0 and 1. This is done as they are not compared, but the trend of their change is visualized. We plot the average values for the training month of February, as a vertical red line, to indicate a "threshold".

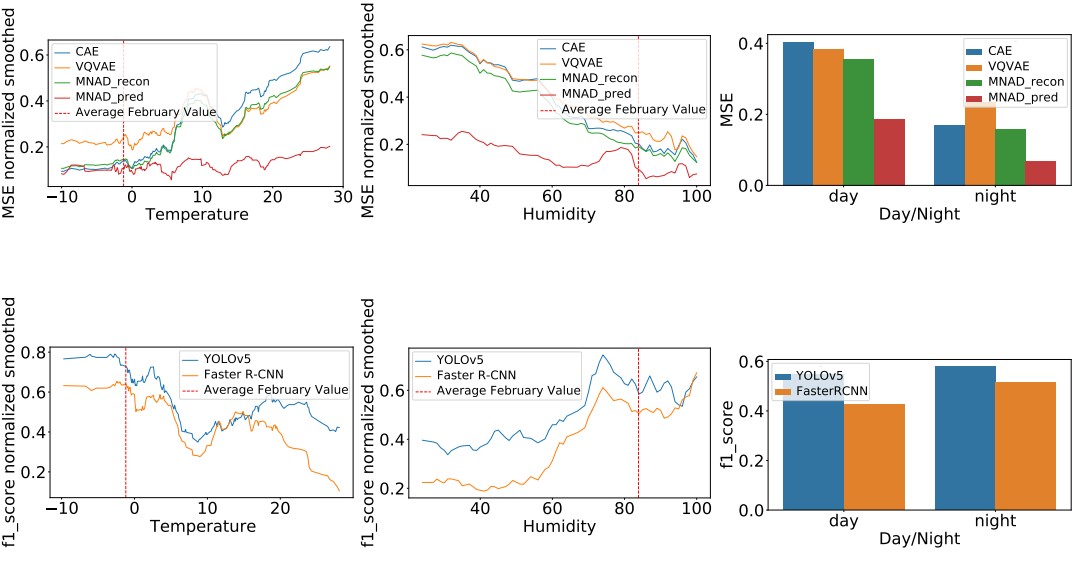

Figure 2: Visual representation of the changes of MSE and F1-score for the tested models compared to the temperature, humidity and day/night cycle.

## 6 Drift Prediction Baseline

As a baseline for exploring and mitigating the effects of concept drift a reference algorithm for predicting drift is presented. We use three strongest features - temperature, humidity and day/night cycle, together with MSE from our convolutional autoencoder (CAE) trained on the February monthly data. The CAE is chosen, as it is the most sensitive to changes in the dataset and is strongly correlated to the performance of all other tested models, except Faster R-CNN. The CAE MSE results from the training data are used together with the chosen features to train two widely used novelty/outlier detection models - isolation forests [42] and one-class SVM [59], available as part of scikit-learn [58]. The isolation forest has 100 base estimators, the one-class SVM has a radial basis function (RBF) kernel and $\gamma$ of 0.03. We then test the results from each day from the full LTD dataset to detect points where many outliers emerge in both predictors. The first large concentration of outliers in 7 consecutive days is selected, which in our case is 5th of March.

To test if taking in consideration data from the found drift point can help with the performance of the models against concept drift, training data from one week starting after the 5th of March is sampled. The new data is used together with the previous training data from February to retrain the tested models. The results, together with the month results from Table 3 and 4 for comparison, are given in Table 7 and Table 8. By adding the March data, all tested models achieve better results. We can see that the outlier detection models trained on the CAE MSE, together with the temperature, humidity and day/night cycle can be used together as a indicator for the amount of drift present in the input data.

## 7 Conclusion and Future Work

In this paper we introduced the Long-term Thermal Drift (LTD) dataset spanning 8 months for detecting concept drift in deep learning models. The dataset and the accompanying metadata can be used to document performance degradation as data drifts from the training set. These effects were studied on anomaly and object detection models, as well as autoencoders. It was demonstrated that more diverse training data lowers the effects of concept drift. The performance of the models showed a strong correlational and causal relationship to the change in temperature and humidity. A less pronounced relationship was observed to the day/night cycle and scene activity. Lastly, we showed

Table 7: The MSE results from the full month in Table 3, compared to the ones using the new training datasets containing a combination of February and the week in March where drift is detected. Higher results show worse performance.

| Methods | Train | Test Jan. | Apr. | Aug. |
|---|---|---|---|---|
| VQVAE2 | Feb. 5k | 0.0021 | 0.0039 | 0.0035 |
| | Feb. 5k + Mar. 5k | **0.0020** | **0.0033** | **0.0030** |
| MNAD Recon. | Feb. 5k | 0.0015 | 0.0041 | 0.0048 |
| | Feb. 5k + Mar. 5k | **0.0006** | **0.0015** | **0.0025** |
| MNAD Pred. | Feb. 5k | 0.0007 | 0.0006 | 0.0007 |
| | Feb. 5k + Mar. 5k | **0.0007** | **0.0005** | **0.0006** |

Table 8: The $mAP_{50}$ Results from the full month in Table 4, compared to the ones using the new training datasets containing a combination of February and the week in March where drift is detected. Lower results show worse performance.

| Method | Train | Test Jan. | Apr. | Aug. |
|---|---|---|---|---|
| YOLOv5 | Feb. 100 | 0.7930 | 0.4860 | 0.4830 |
| | Feb. 100 + Mar. 100 | **0.8690** | **0.6640** | **0.6110** |
| Faster R-CNN | Feb. 100 | 0.6400 | 0.2560 | 0.3180 |
| | Feb. 100 + Mar. 100 | **0.6990** | **0.3910** | **0.3380** |

how the concept drift can be further mitigated by detecting when it starts to manifest and providing additional data to the training process.

The proposed LTD dataset contains a combination of diverse environmental images and granular metadata. The equally spaced long-term data can be used to test the change in performance of deep learning models at different data scenarios - only day or night data, changes between activity in the weekday and weekends, summer and winter scenarios. The influence of weather conditions like rain, snow or fog can also be explored. The possibility of training more robust models and predicting when steps need to be taken, before their performance degrades, is only possible with such long-term sequential datasets.

Possible negative social impacts of such long-term datasets concentrating on a single location is that they can be used to track the habits, interactions and movements of people. We offset this by providing a thermal dataset, which provides greater protection of people's anonymity than conventional RGB and does not require post-processing for blurring facial features.

The long-term nature of the dataset can also be used, as demonstrated in this paper, to utilize time-series analysis procedures on the outputs from different layers of deep learning models. From simple time-series analysis and forecasting models like Vector Autoregressive (VAR) Models [29] to more complex and data agnostic models like STRIPE [25] or Adversarial Sparse Transformers [84].

We believe that the proposed dataset and the accompanied analysis would help researchers understand the causes for performance drift in models and hence enable easier deployment of long-term solutions in outdoor environments.

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
