# OpenReview forum: "Seasons in Drift: A Long Term Thermal Imaging Dataset for Studying Concept Drift"
_NeurIPS.cc/2021/Track/Datasets_and_Benchmarks/Round2 — NeurIPS 2021 Datasets and Benchmarks Track (Round 2)_

### Official Review · Reviewer_ZaV9 · 2021-09-20
**A dataset to study the role of seasonality, time of year and day distribution drift on model performance**

**Rating:** 7
**Confidence:** 4
**Correctness:** The claims of the paper are mainly co…

**Strengths:**

-- The paper presents a novel thermal imaging dataset spanning many months. The dataset contains metadata that allows researchers to study the correlations between meta-features and model performance.
-- The paper also has some bounding box labels allowing for some analysis of relationships between task diversity and concept drift.
-- The paper performance a few benchmarks on the correlation between concept drift and model performance.

**Weaknesses:**

-- Due to the limited amounts of object detection data, this dataset will only enable model-centric methods to handle concept drift instead of data-centric methods.
-- The limited scene diversity also limits the dataset to studies of data distribution shifts caused by weather, day, and time of year variations.

**Additional Feedback:**

none.

**Clarity:**

The paper is well-written and straightforward. However, some aspects of the presentation can improve. Specifically, the labeling and subset information can be moved out of the experiments section and added to the dataset section.

**Documentation:**

The dataset and meta-data are well documented, but the model training code/configs appear to be missing.

**Ethics:**

none.

**Relation To Prior Work:**

The references to prior work appear adequate.

**Summary And Contributions:**

In this paper, the authors present a novel dataset to study the effect of data distribution drift on model performance. The dataset consists of thermal video taken across months, metadata about the weather, day/night cycle, etc. The authors benchmark many model types to understand the effect of time of year data drift on the model performance. Lastly, the authors study the correlation between model performance and metadata.

The paper's main contributions are a novel thermal imaging dataset taken across many months. The datasets come with metadata about the world state during the video capture. Lastly, the dataset comes with some bounding box labels for object detection tests.

---

> ### Author Response · Authors · 2021-09-27
> **Response to Reviewer #3 (ZaV9)**
>
> The authors wish to thank the reviewer for their feedback and their belief in the strengths of the paper. We would like to address some of the concerns of the reviewer below.
>
> - **Due to the limited amounts of object detection data, this dataset will only enable model-centric methods to handle concept drift instead of data-centric methods.**
>
> The annotation of the entire dataset has already begun since August, and we expect to have the entire number of frames of 6750 2-min video clips annotated by November. We will then update the repository of the dataset with the new annotation. This would give a wider possibility of use cases around object detection, tracking, etc.
>
> - **The limited scene diversity also limits the dataset to studies of data distribution shifts caused by weather, day, and time of year variations.**
>
> We agree that the best-case scenario would be having multiple locations with long-term single-scene data. This would ensure that both enough diverse data would be captured for better generalization, but also the effects of the background would be minimized. Having said that, we believe that the value of a long-term dataset containing environmental and time metadata is still quite big. The ease of sampling of specific data can give researchers the possibility to “mix and match” subsets depending on their needs. We also hope that this dataset would give other researchers the push to create more single location long-term datasets with metadata, which can be used in combination with this one.
>
> - **The dataset and meta-data are well documented, but the model training code/configs appear to be missing.**
>
> The model training code and config files can be found in the Methods sub-directory in the GitHub repository - [https://github.com/IvanNik17/Seasonal-Changes-in-Thermal-Surveillance-Imaging/tree/main/methods](https://github.com/IvanNik17/Seasonal-Changes-in-Thermal-Surveillance-Imaging/tree/main/methods). To make it more intuitive to find these we have updated the GitHub repository ReadMe to better reflect a map of the repo and added missing run commands.

---

### Official Review · Reviewer_WNzq · 2021-09-21
**Seasons in Drift: Intent to provide a thermal imaging dataset across 8 months**

**Rating:** 6
**Confidence:** 4
**Clarity:** Yes, in terms of language. Minor edit…

**Strengths:**

Consistent dataset across 8 months along with the metadata.

**Weaknesses:**

The literature review feels very short and incomplete as it does not account for many representative datasets in robotics that have been studied the drift concept for many years already: BDD 100k, Oxford Robot Car Seasons, CMU Seasons, DENSE, Ford Multi-AV Seasonal, FLIR dataset, etc.
More comments about how the data format is would be appreciated. For example: how does the information from the IR sensor compare to RGB? is it an 8-bit grayscale image with corresponding intensity values for temperatures?
No comments on how the algorithms used are adapted to IR images are given. It is briefly mentioned that the object detectors used have already been tested on thermal images. Is this the same case for the anomaly detection and the outlier detection methods? If they already have been used previously, please provide a brief description or intuition of what changes or adaptations were necessary for the algorithms to work.
Line 204-206: This needs to be reworded either to make it a hypotheses to be proven or as a conclusion from their result.
line 218: 100 frames is incredibly small for such network models for any significant learning and the pretraining details are missing.
Line 239-241: What about camera movement from wind? That would make the mean value of the difference ineffective as a measure of activity in the scene.
Some analysis ensuring no overfitting of the models is occurring would help to strengthen the effectiveness.
Last paragraph of the conclusion: It is stated that the proposed dataset can help researchers understand the causes of drift. But with the lack of a metric of how much drift is occurring this exercise will be half successful. Some indication of the quantity of drift will be helpful.
Appex Dataset supp material: Section 1.5: This cannot be used for effective tracking due to the frequency of the data collected and long term nature of data. (2 min clips)


**Additional Feedback:**

N/A

**Correctness:**

There is a consistent process with regards to timing, frequency of the data collected. That seems sound, but lacks on details as mentioned above in weaknesses to show the completeness in the process. The Kaggle page also lacked in terms of details.



**Documentation:**

There is some information provided but falls short on details.

**Ethics:**

Overall given that this is camera surveillance on a harbor, there is reduced chances of personal identifiable information in addition to the data being thermal. That said, some question in the review should be addressed like: Did you discuss any potential negative societal impacts of your work?

**Relation To Prior Work:**

See weaknesses.

**Summary And Contributions:**

The paper introduces a new thermal data set for video surveillance focusing on studying how ML algorithms can be affected by several environmental factors (aka as drift) when tested across multiple seasons (for about eight months). Building machine learning models resilient to drift is paramount for deploying and functioning persistently outside Petri-dishes use cases.
The authors evaluate auto-encoders, object detection, and anomaly detection algorithms for this dataset, highlighting drift effects.

---

> ### Author Response · Authors · 2021-09-27
> **Response to Reviewer #2 (WNzq)**
>
> We appreciate the in-depth evaluation and the provided valuable pointers to making the proposed article better. We would like to address the concerns raised in the review.
>
> - **The literature review feels very short and incomplete**
>
> The prior work was shortened because of lack of space, omitting datasets that should have been left in. We will use the additional page to expand the literature review and add the datasets pointed out by the reviewer, as well as additional ones. We have reorganized them to better present the topic, where our dataset fits, and what problems it addresses. We have rewritten Table 1 to better represent the prior work.
>
> - **More comments about how the data format is would be appreciated**
>
> We have added additional explanation as part of Section 3 in the paper, as well as a better overview of the formatting to the Kaggle dataset description. Both for the gather images and the captured metadata and their structures.
>
> - **How does the information from the IR sensor compare to RGB? is it an 8-bit grayscale image with corresponding intensity values for temperatures?**
>
> The IR sensor on the Hikvision DS-2TD2235D-25/50 is a long-wavelength infrared (LWIR) unit, capturing wavelengths between 8 and 14 micrometers. It produces 8-bit uncalibrated grayscale images. We have added this information to the paper.
>
> - **No comments on how the algorithms used are adapted to IR images are given.**
>
> The only change to the used autoencoder and object detector methods is the reduction of the input channel number from 3 to 1, corresponding to a change from RGB to Grayscale input. In our paper getting high algorithm performance is not the main focus, so we adapted the methods to work with our thermal dataset. We have added this information to the paper.
>
> - **Line 204-206: This needs to be reworded either to make it a hypotheses to be proven or as a conclusion from their result**
>
> We have moved this statement to line 228-229 and reworked it as a conclusion, as the findings show that using data from a longer period of time for training gives better results.
>
> - **Line 218: 100 frames is incredibly small for such network models for any significant learning and the pretraining details are missing**
>
> We agree that the 100 frames is a very small amount and that better and more robust results would have been achieved with a much larger training and validation set. We address this by using a pre-trained model. We have added more information on the number of images used for pre-training and the necessary changes made for that. We monitored the validation loss and empirically chose the amount of training epochs to prevent overfitting to the relatively small amount of data. We have added this clarification to the methods overview part of the paper.
>
> - **What about camera movement from wind? That would make the mean value of the difference ineffective as a measure of activity in the scene.**
>
> To mitigate any unwanted movement influencing the difference between images, we have masked everything in the background and foreground that might be influenced by wind and camera sway. We extracted a large number of the dataset difference images and manually checked them and have seen that this approach prevents unwanted movement and even though there are still some artifacts and noise present, they are negligible and are offset by the simplicity of our approach.
>
> - **Some indication of the quantity of drift will be helpful.**
>
> We believe that the use of the combination between the CAE reconstruction error and the changes of the proposed metadata like temperature, humidity, and day/night cycle can be a useful indicator for the amount of drift present. These can be used as initial indicators, which can be further built upon by other autoencoders or statistical time series methods. We have emphasized the use of these metrics as indicators for drift as part of the conclusion.
>
> - **This cannot be used for effective tracking due to the frequency of the data collected and long term nature of data**
>
> We agree with this. We plan to add not only the 2-minute clip dataset, but also the raw video dataset as part of the publication, together with all the necessary code for processing the data (as indicated in Section 1.6 in the Appendix). This way, if needed, longer clips or different sampling frequencies can be employed and used for other types of vision tasks. In addition, we plan to annotate a large part of the dataset by November and update the Kaggle repository, making it even more useful for different supervised learning tests.
>
> - **The Kaggle page also lacked in terms of details.**
>
> We agree that more information is needed and we have added more details to the Kaggle page about the used hardware, the data pre-processing, formatting, and sampling.

---

### Official Review · Reviewer_1K6a · 2021-09-23
**Useful thermal imaging dataset that can enable deployment of long-term solutions robust to concept drift**

**Rating:** 6
**Confidence:** 3
**Clarity:** Yes, the paper is clear and well writ…

**Strengths:**

- Makes a long-term public thermal dataset accessible for the community. Thermal imaging datasets enable personal data to be protected without requiring extensive post-processing, unlike RGB datasets. Hence, this dataset can comply with GDPR.
- Large dataset almost twice the size of the other available thermal datasets (Table 1).
- Metadata analysis is well-motivated in light of the application niche of the dataset.
- Recent deep methods have been used for the baseline experiments and the experiments are appropriately motivated.
- The drift analysis of the performance of methods compared with environmental factors is sufficient.


**Weaknesses:**

- Description of prior art seems to deliberately exclude Domain Adaptation works. DA, especially for applications in autonomous driving deal with season shifts and relevant 'concept' shifts such as time-of-day and day-to-night and widely different scenario shifts and methods, have performed well for detection and pixel-level labeling tasks. They also use real-world data such as [1] and [2]. Authors should address why and how these datasets are/are not relevant to the seasonal shifts discussed.
- Similarly, [1] and [3] also include additional metadata such as geographic location and scenes of the same location captured at various time/seasonal intervals. This enables datasets to be balanced, whereas capturing long-term videos with concept shifts at very few time intervals may lead to imbalanced datasets that may not generalize well to different geographical locations. These are relevant to be discussed here especially in light of why and what metadata is useful for concept shift identification.

References:
[1] Sakaridis, C., Dai, D. and Van Gool, L., 2021. ACDC: The Adverse Conditions Dataset with Correspondences for Semantic Driving Scene Understanding. arXiv preprint arXiv:2104.13395.

[2] Yu, F., Chen, H., Wang, X., Xian, W., Chen, Y., Liu, F., Madhavan, V. and Darrell, T., 2020. Bdd100k: A diverse driving dataset for heterogeneous multitask learning. In Proceedings of the IEEE/CVF conference on computer vision and pattern recognition (pp. 2636-2645).

[3] Sakaridis, C., Dai, D. and Gool, L.V., 2019. Guided curriculum model adaptation and uncertainty-aware evaluation for semantic nighttime image segmentation. In Proceedings of the IEEE/CVF International Conference on Computer Vision (pp. 7374-7383).


**Additional Feedback:**

L:106 imagining -> imaging

**Correctness:**

Yes, the submission is a dataset constructed in a manner useful for the machine learning community. The evaluation methods and experiments are appropriate. The dataset and the code to reproduce the experiments were shared.

**Documentation:**

There is sufficient detail on the dataset and how the dataset was collected. The paper includes documentation and intended uses cases for the dataset. The dataset is uploaded on Kaggle.

**Ethics:**

There seem to be no ethical concerns that warrant further discussions. However, the wordings in L:106-108 are generic and may need further review. The highlighted words in: "However, thermal imagining **generally makes it easier** for personal data to be protected and **reduces the requirement** for additional post-processing to ensure person anonymity, which is a crucial requirement for complying with the European general data protection regulations (GDPR)." The explanation in Checklist 1(d) is seemingly enough.

**Relation To Prior Work:**

Please refer to the Weaknesses section. Relation to prior work could include more discussions on datasets and methods that deal with similar niche outcomes albeit for a different use case (DA methods for example).

**Summary And Contributions:**

- Introduces and makes available a long-term thermal dataset for investigating gradual concept drift. The dataset also includes additional metadata (timestamp and weather) that may be useful for a variety of associated tasks.
- Analysis of the causal relationship and correlations between the environment and performance.
- Performance of recent methods has been studied (Sec. 4.3) and analysis of performance compared with environmental variables (Sec. 5) is also analyzed.

---

> ### Author Response · Authors · 2021-09-27
> **Response to Reviewer #1 (1K6a)**
>
> Thank you for your thoughtful review and given insights, as well as the suggested related work. We would like to address your suggestions and comments below:
>
> - **Description of prior art seems to deliberately exclude Domain Adaptation works.**
>
> Thank you for pointing out the omission of Domain Adaptation self-driving car work from our prior work overview. We see the concept drift problem represented in two types of long-term video use cases – ones that have a changing location and background environment like self-driving cars, robotics or human egocentric footage and ones that have constant location footage like surveillance and CCTV camera videos. We agree that both are relevant and comparing their current state is important, so we expanded the prior work to also represent datasets better for scenarios with changing locations and focused on how Domain Adaptation work fits in with the problem.
>
> - **Capturing long-term videos with concept shifts at very few time intervals may lead to imbalanced datasets**
>
> We have added the datasets mentioned by the reviewer to the dataset overview Table 1, together with their provided metadata to ensure that readers can if necessary, compare and combine them.
> We believe that the provided metadata can be very useful in selecting parts of the dataset that contain specific concept shifts. It should be straightforward to select specific days, weeks, months from a certain season or directly query the weather data to include videos containing video footage from different temperatures, humidity, sun radiation, etc. to create subsets for training and testing. Of course, the best solution would be to have a long-term fixed position dataset from multiple locations, to ensure better generalization. Even as such, we believe that our LTD dataset can be useful in multi-dataset domain generalization problems, because of its diversity. This paper can also inspire others to both add diverse metadata for the environmental conditions and times of the captured data, as well as focus on longer-term data.
>
> - **106-108 are generic and may need further review**
>
> We agree with your statement and have rewritten the sentences to better explain the benefits of the proposed thermal dataset and to make it less generic and prone to misinterpretation.

---

### Decision · Program_Chairs · 2021-10-09

**Decision:**

Accept

**Comment:**

All reviewers recommend acceptance. Congratulations to authors! I urge the authors to incorporate reviewers' suggestion in the final version of the paper.